# Seasonal variations in gut microbiota and disease course in patients with inflammatory bowel disease

**Mizuki Tani**[1], **Shinichiro Shinzaki**[1], **Akiko Asakura**[1], **Taku Tashiro**[1], **Takahiro Amano**[1], **Yuriko Otake-Kasamoto**[1], **Takeo Yoshihara**[1], **Shunsuke Yoshii**[1], **Yoshiki Tsujii**[1], **Yoshito Hayashi**[1], **Takahiro Inoue**[1], **Daisuke Motooka**[2], **Shota Nakamura**[2], **Hideki Iijima**[1,3], **Tetsuo Takehara**[1] *

1 Department of Gastroenterology and Hepatology, Graduate School of Medicine, Osaka University, Suita, Japan, 2 Genome Information Research Centre, Research Institute for Microbial Diseases, Osaka University, Suita, Osaka, 3 Department of Internal Medicine, Osaka Police Hospital, Osaka, Japan

* takehara@gh.med.osaka-u.ac.jp

**Data Availability Statement:** All relevant data are within the paper and its Supporting Information files. The metagenome sequencing datasets are

## Abstract

### Background and aim

Environmental factors are associated with onset and course of inflammatory bowel disease (IBD). Our previous study by about 1,100 IBD patients revealed half of the patients experienced seasonal exacerbation of disease. We investigated the seasonality of fecal microbiota composition of IBD patients.

### Methods

Fecal samples were consecutively collected in each season from IBD outpatients and healthy controls between November 2015 and April 2019. Participants who were treated with full elemental diet or antibiotics within 6 months or had ostomates were excluded. Bacterial profiles were analyzed by 16S rRNA sequencing, and the changes between the diseases and seasons were compared.

### Results

A total of 188 fecal samples were analyzed from 47 participants comprising 19 Crohn's disease (CD) patients, 20 ulcerative colitis (UC) patients, and 8 healthy controls (HC). In CD patients, the phylum *Actinobacteria* and *TM7* were both significantly more abundant in autumn than in spring and winter, but not in UC patients and HC. Moreover, the genera *Actinomyces*, a member of *Actinobacteria*, and *c_TM7-3;o_;f_;g_ (TM7-3)*, that of *TM7*, were significantly more abundant in autumn than in spring, and the abundance of *Actinomyces* was significantly correlated with that of *TM7-3* throughout the year in CD patients, but not in UC patients and HC. CD patients with high abundance of *TM7-3* in the autumn required significantly fewer therapeutic intervention than those without seasonal fluctuation.

available in the NCBI/EMBL/DDBJ database (PRJNA854295).

**Funding:** This work was funded by the Japan Society for the Promotion of Science KAKENHI grant numbers 17H04159 and 19K08394. The funders had no role in study design, data collection and analysis, decision to publish, or preparation of the manuscript.

**Competing interests:** The authors have declared that no competing interests exist.

## Conclusions

Oral commensals *Actinomyces* and its symbiont *TM7-3* were correlatively fluctuated in the feces of CD patients by season, which could affect the disease course.

## Introduction

The number of patients with inflammatory bowel disease (IBD) increased in Europe and North America during the second half of the 20th century, while the number of patients plateaued in the 21st century. The number of IBD patients has increased significantly in Asian countries such as Japan and India, which were previously thought to be low-risk areas for disease onset. This dramatic short-term change cannot be explained by genetic alterations alone, and other factors, such as environmental factors, are considered to be associated with disease exacerbation [1,2]. There is also increasing evidence that the intestinal microbiota is involved in the pathogenicity of IBD. Dysbiosis in IBD is commonly characterized by an increase in the abundance of *Proteobacteria* and a decrease in the abundance of *Firmicutes*, along with a decrease in constituent bacterial richness [3,4]. Fecal microbiota transplantation (FMT), targeted at altering the intestinal microbiota, is drawing attention as a new treatment in patients with ulcerative colitis (UC), and dysbiosis has been observed in the small bowel of colonic Crohn's disease (CD) patients [5,6]. However, the relationship between environmental factors and the fecal microbiota composition is still unclear.

We recently conducted a survey of more than 1,000 IBD outpatients and reported that half of the IBD patients noted seasonality of disease exacerbation, especially in winter [7]. Although the intestinal microbiota is generally stable throughout the year in healthy people, the fecal microbiota composition can vary according to race and residence country [8]. A recent report from Europe showed that in IBD patients, a poor microbiota composition was related to decreased serum 25(OH)D levels in winter and spring [9]. However, it is still unclear whether the intestinal microbiota is seasonally altered in IBD patients. Among the environmental factors that influence IBD disease, we considered the intestinal microbiota as one of the factors that change seasonally. Therefore, we conducted an exploratory study to determine whether seasonal changes in the fecal microbiota are observed in IBD patients and, if so, whether they are associated with seasonal disease exacerbations.

## Methods

### Participants

This prospective observational study was conducted between November 2015 and April 2019. Participants were 39 IBD outpatients (20 UC patients and 19 CD patients) at Osaka University Hospital and 8 volunteer healthy controls (HCs). Patients were excluded if they were less than 18 years old, treated with a full elemental diet, treated with antibiotics within 6 the previous months or had undergone ostomy surgery. CD and UC were diagnosed based on established criteria, and the disease activity of CD and UC was assessed using the Crohn's disease activity index (CDAI), and the partial Mayo score (PMS), respectively [10–12]. The study protocol was approved by the ethics committee of Osaka University Hospital (No.15174) and conducted in accordance with the latest version of the Declaration of Helsinki. Prior to the study, written informed consent was obtained from all participants. For non-microbiome data, the authors

had access to information that could identify individual participants during or after data collection in their usual medical care.

## Study design

Participants were asked to collect fecal samples themselves and fill out questionnaires to determine disease activity in four consecutive seasons. March-May was defined as spring, June-August as summer, September-November as autumn, and December-February as winter. The earliest date of each seasonal outpatient visit was set as the specimen collection date. The fecal samples were placed into disposable sterile feces tubes (20 ml; Watson, Kobe, Japan) by the patients and then preserved immediately at -80 degrees centigrade until DNA extraction.

Certain bacteria that showed seasonal change were examined for correlation with a one-year clinical activity index as a short-term course. Divided into two groups based on the amount of seasonal change, long-term clinical outcome was evaluated by investigating the time to therapeutic intervention during 3 years after specimen collection. Therapeutic intervention was defined as additional medical/surgical treatment, endoscopic dilation for intestinal stenosis, and hospitalization due to exacerbation of the disease.

## Bioinformatic processing

Bacterial DNA was extracted from the fecal samples by using a DNeasy PowerSoil Kit (Qiagen, Hilden, Germany) according to the manufacturer's protocol. We sequenced the V1-V2 variable region of the 16S rRNA gene on the Illumina MiSeq platform (Illumina, San Diego, CA, USA) with 251-bp paired-end sequencing. The paired-end sequences obtained were merged, filtered, and denoised using DADA2 [13]. Taxomic assignment was performed using the QIIME2 feature-classifier plugin with the Greengenes 13_8 database. The QIIME2 pipeline, version 2020.2, was used as the bioinformatics environment for the processing of all relevant raw sequencing data (https://qiime2.org).

## Statistical analysis

All statistical analyzes were performed using JMP pro version 15.2.1 (SAS Institute, Cary, NC, USA). The relative abundance of bacterial genera among the disease states and the UniFrac distance obtained by principal coordinates analysis (PCoA) were evaluated using Tukey's honestly significant difference (HSD) test. Pearson's test was performed to analyze the association with bacterial abundance. In these analyzes, $p < 0.05$ was considered statistically significant. Comparisons of the alpha-diversity and bacterial richness among seasons were performed by Wilcoxon's signed rank test with a corresponding paired test. We divided 0.05 by 6 for the comparison between each season (four groups) and considered $p < 0.0083$ to be statistically significant (Bonferroni adjustment as a post hoc test). Kaplan–Meier curves were used to analyze the cumulative incidence of therapeutic intervention between the groups with log-rank tests.

## Results

### Participant characteristics

A total of 47 participants, including 39 IBD patients and 8 HCs, were enrolled in this study. All participants were Japanese, and their clinical characteristics at recruitment are summarized in Table 1. The median (range) ages of CD and UC patients and HCs were 44 (32–55), 47 (44–57), and 33 (32–43) years, respectively. The median (range) activity score was 83 (59–123) in CD (CDAI) and 0 (0–1) in UC patients (PMS). Because the participants in the HC group were

**Table 1.  Baseline characteristics of the study population.**

| | CD n = 19 | UC n = 20 | HC n = 8 | *P* value |
|---|---|---|---|---|
| Sex, male, n (%) | 14 (73.6) | 13 (65.0) | 6 (75.0) | 0.9138 |
| Age at enrollment, y/o, median (range) | 44 (32–55) | 47 (44–57) | 33 (32–43) | 0.0753 |
| Age at diagnosis, y/o, median (range) | 26 (21–33) | 40 (29–48) | – | |
| BMI, kg/m$^2$, median (range) | 19.4 (18.4–22.9) | 22.1 (19.9–23.6) | 22 (20.4–23.8) | 0.1224 |
| Current smoking, n (%) | 2 (11%) | 4 (20%) | 0 (0%) | 0.5300 |
| Alcohol use, n (%) | 4 (21%) | 7 (35%) | 2 (25%) | 0.7548 |
| Location,n (%) | | | | |
| ileum/colon/ileocolon/upper (CD) | 7/5/7/1 | | | |
| pancolitis/left-sided/rectum (UC) | | 10/5/5 | | |
| Disease activity, median (range) | CDAI 83 (59–123) | PMS 0 (0–1) | – | |
| Treatment | | | | |
| 5-ASA/Bio$^#$/IM | 19/9/2 | 17/2/8 | 0 | |
| PPI/Probiotics/PSL | 4/14/1 | 1/16/0 | 0 | |
| Gastrointestinal surgery, n (%) | 10 (52.6) | 2 (10) | 0(0) | 0.002 |

Bio; Biologics, IM; immunomodulator, PPI; proton pump inhibitor.

$^#$CD: IFX (Infliximab) n = 5, ADA (Adalimumab) n = 3, UST (Ustekinumab) n = 1, UC:IFX n = 2, ADA n = 0, UST n = 0.

volunteers, it was difficult to adjust the characteristics of the HC group with the patient group. We could match the proportion of sex; however, the HC group tended to be younger compared to the UC and CD groups, but not significant. Only the surgical rate was significantly different between the CD group an UC group.

## Diversity of fecal microbiota

We first analyzed 188 fecal samples collected during four seasons from 47 participants and evaluated the diversity of the gut microbiota using alpha-diversity indices. The Shannon index, an indicator of microbial community diversity, was significantly lower in IBD patients than in HCs and lower in CD patients than in UC patients among the IBD patients (**Fig 1A**) [14]. These indices did not change throughout the four seasons in either UC or CD patients. Stable alpha-diversity among the seasons was also observed in the HCs (**S1 Fig**).

The overall structures of the gut microbiomes in IBD patients and HCs using beta-diversity indices were calculated with weighted UniFrac distances. The three principal components (PC1, PC2, and PC3) are shown in **Fig 1B**. PCoA revealed that both CD and UC clusters were located apart from HC clusters. Moreover, the UniFrac distance was significantly greater in CD patients than in UC patients and significantly greater than that in HCs for both diseases (**Fig 1C**).

## Seasonal changes in the fecal microbiota in IBD patients and HCs at the phylum level

We analyzed the fecal microbiota abundance at the phylum level. The abundances of *Proteobacteria* and *TM7* in CD patients in the four seasons were significantly higher than those in UC patients and HCs (**S2A Fig**). We next analyzed the abundances of the top eight phyla in each season by disease. In the HCs, there were no intraseasonal differences in these bacterial groups (**S2B Fig**). In CD patients, however, the phyla *Actinobacteria* and *TM7* were significantly more abundant in autumn than in spring and winter But *Actinobacteria* and *TM7* in autumn did not differ significantly in summer. (**Fig 2A**). In UC patients, although

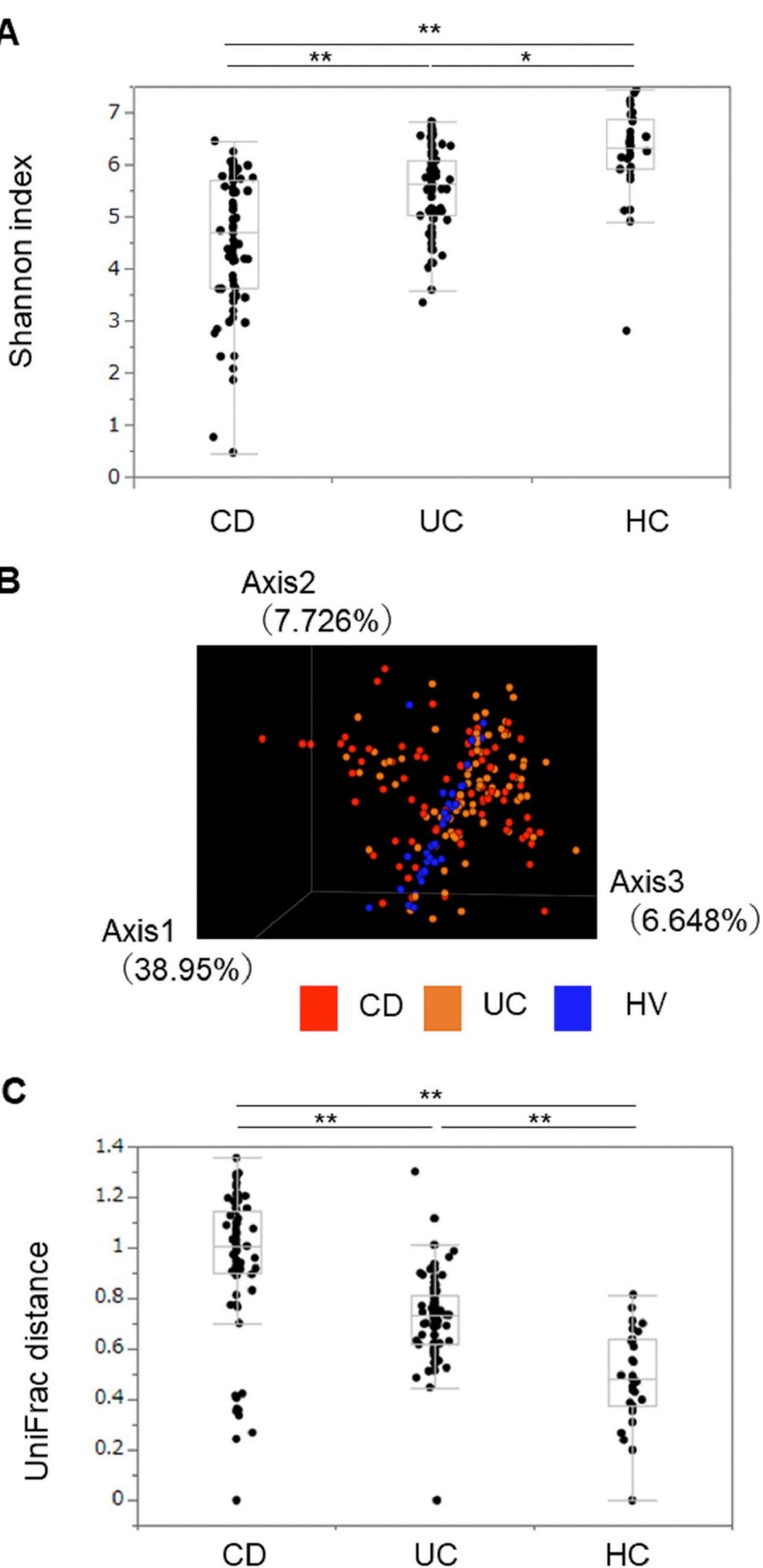

**Fig 1. Fecal bacterial diversity in the inflammatory bowel disease (IBD) patients and Healthy Controls (HCs).** (A) **Alpha-diversity (Shannon index) in the IBD patients and HCs.** Tukey–Kramer HSD test. * indicates p <0.005, ** indicates p < 0.0001. (B) **Beta-diversity in the IBD patients and HCs.** All the specimens of the target patients are represented in a three-dimensional principal coordinate analysis. The red circles represent the CD patients, the orange circles represent the UC patients, and the blue circles represent the HCs. Each data point represents an individual sample. (C) **UniFrac distances in IBD patients and HCs.** A p value of less than 0.05 was considered significant. ** indicates p < 0.0001.

*Actinobacteria* was significantly more abundant in autumn than in spring, *TM7* remained unchanged throughout the season (**Fig 2B**). No intraseasonal differences were observed in the other bacterial groups.

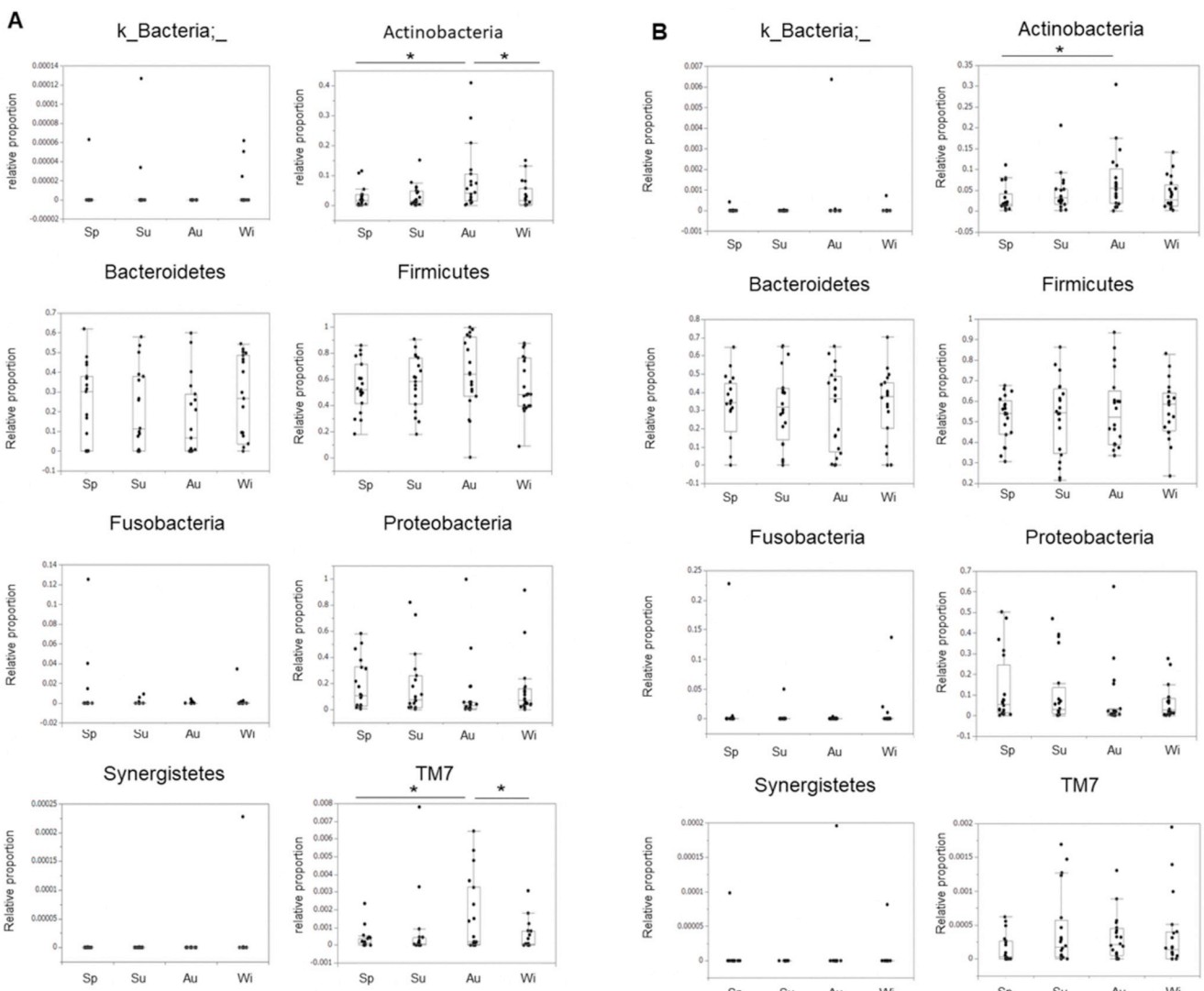

**Fig 2. Seasonal changes in bacterial abundance in the IBD patients (phylum level).** In the CD patients (A) and UC patients (B), the eight main phyla accounting for the largest proportions were examined. Phyla are listed in alphabetical order. The vertical axis shows the relative proportions, and the horizontal axis shows the season (*Sp: Spring, Su: Summer, Au: Autumn, Wi: Winter). Bonferroni correction was performed with a paired test for correspondence, and a p value of less than 0.0083 was considered significant. * indicates the p value of less than p < 0.0083.

## Seasonal changes in *Actinobacteria* and *TM7* in IBD patients and HCs at the genus level

We next focused on the phyla *Actinobacteria* and *TM7* and investigated seasonal variations at the genus level. For *Actinobacteria*, in the CD patients, *Actinomyces* was significantly more abundant in autumn than in spring. In the UC patients and HCs, there were no seasonal fluctuations in these bacterial groups (**Fig 3**).

For *TM7*, in the CD patients, a seasonal change in *c_TM7-3;o_;f_;g_ (TM7-3)*, which was more abundant in autumn than in spring and in winter, was observed. There were no seasonal changes in the bacterial groups, including *TM7-3*, in the UC patients and HCs (**Fig 4**).

When comparing the abundance of these bacteria in the four seasons between the disease groups, *Actinomyces* was significantly more prevalent in the CD patients than in the UC patients, and *Bifidobacterium* and *TM7-3* were significantly more prevalent in the IBD patients than in the HCs (**S3 Fig**).

## Correlation between *Actinobacteria* and *TM7*

Both *Actinomyces* and *TM7* are oral commensal bacteria, and *TM7* grows on the surface of its host bacterial species *Actinomyces odontolyticus* strain (XH001), a member of the phylum *Actinobacteria* [15]. We investigated the correlation between *Actinobacteria* and *TM7* abundances based on the hypothesis that the abundance of *Actinobacteria* is linked to that of *TM7*. At the phylum level, the abundance of *Actinobacteria* did not significantly correlate with that of *TM7* in either the IBD patients or HCs (**S4 Fig**). However, at the genus level, *Actinomyces* was significantly correlated with *TM7-3* in the CD patients, with a correlation coefficient of 0.7739 ($p < 0.0001$, **Fig 5**). A significant correlation was observed between *Actinomyces* and *TM7-3* throughout the four seasons in the CD patients (**S5A Fig**). However, no correlation with these bacteria was observed in any season in the UC patients (**S5B Fig**). A weak correlation between *Actinomyces* and *TM7-3* was observed in the HCs, with a correlation coefficient of 0.6024 ($p = 0.0003$, **Fig 5**), and the correlations were significant in autumn and winter (**S5C Fig**).

## Short and long-term outcomes in CD patients

Finally, we examined whether seasonal variations in fecal microbiota affect disease activity. In short term, *Actinomyces* and *TM7-3* are bacteria that showed seasonal variation in CD patients, but there was no correlation with CDAI in both bacteria (**S6 Fig**). And there were also no seasonal differences in CRP and clinical activity indices during the four consecutive seasons (**S7 Fig**). We then performed long-term analysis by dividing patients into two groups by the median value of the difference of abundance in *Actinomyces* or *TM7-3* in the autumn compared with that in the spring. Although some CD patients had high CDAI, all of them were outpatients who were clinically stable and did not require any therapeutic intervention during the first year such as additional medical/surgical treatment, endoscopic dilation for intestinal stenosis, and hospitalization. Although no significant changes were observed in *Actinomyces*, the patients with high abundance of *TM7-3* in the autumn than in the spring required significantly fewer therapeutic intervention for 3 years than those with low abundance (**Fig 6**).

## Discussion

In the present study, we clearly showed that the abundances of *Actinobacteria* and *TM7* were significantly higher in autumn than in spring and winter, and the seasonal differences were most pronounced in CD patients. In contrast, in UC patients, only *Actinobacteria* was significantly more abundant in autumn than in spring. We also showed that the fecal microbiota

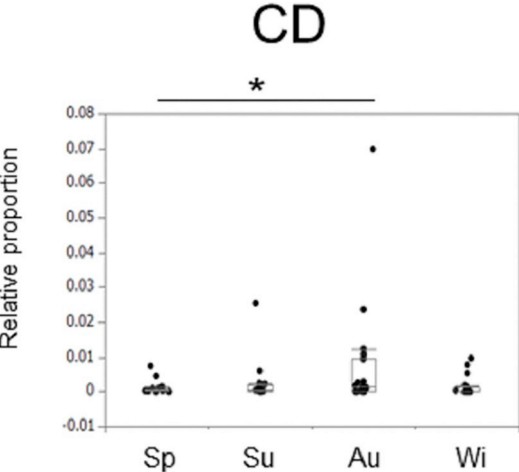

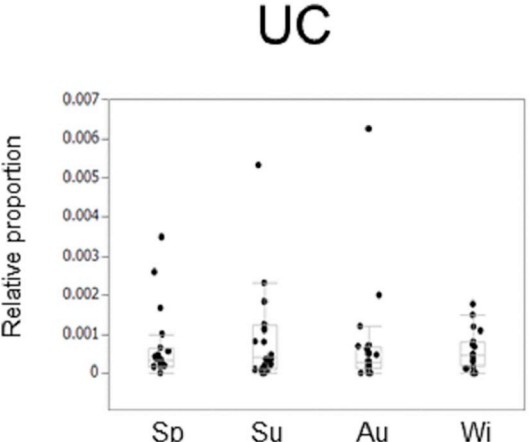

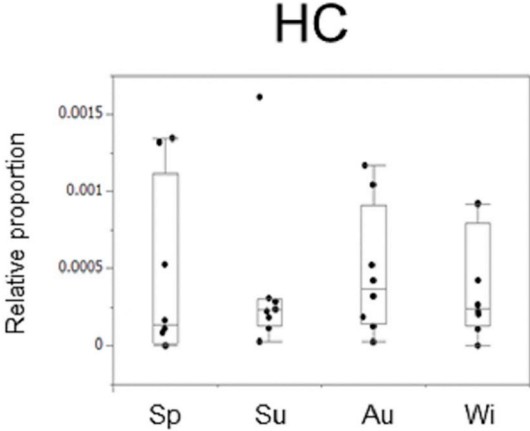

**Fig 3. Seasonal changes in the *Actinomyces* in the IBD patients.** Bonferroni correction was performed with a paired test for correspondence, and a p value of less than 0.0083 was considered significant. * indicates p < 0.0083.

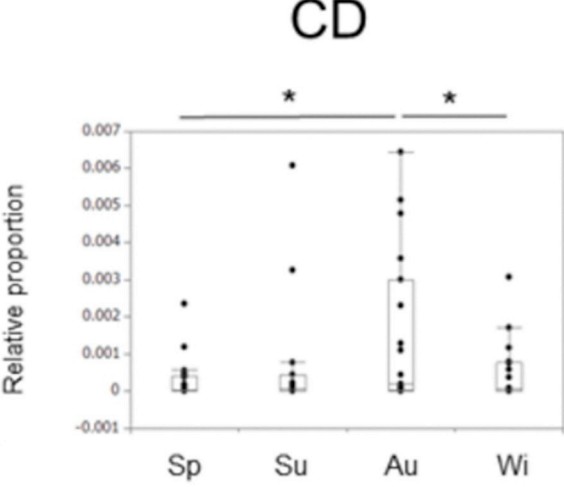

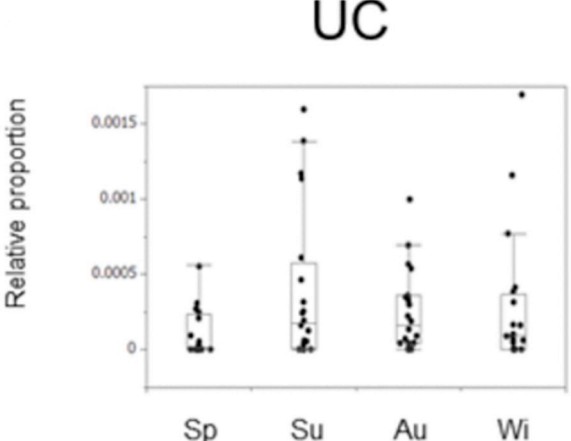

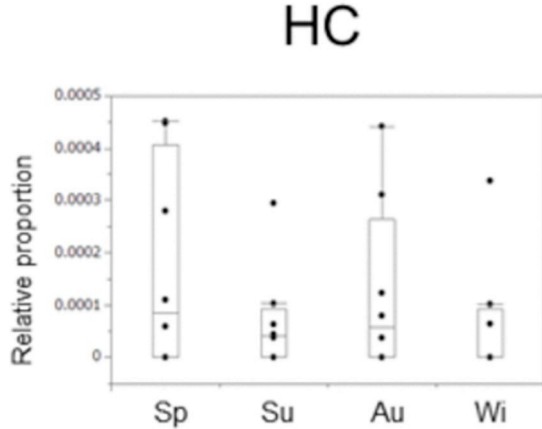

**Fig 4. Seasonal changes in the *TM7-3* in the IBD patients.** Bonferroni correction was performed with a paired test for correspondence, and a p value of less than 0.0083 was considered significant. * indicates p < 0.0083.

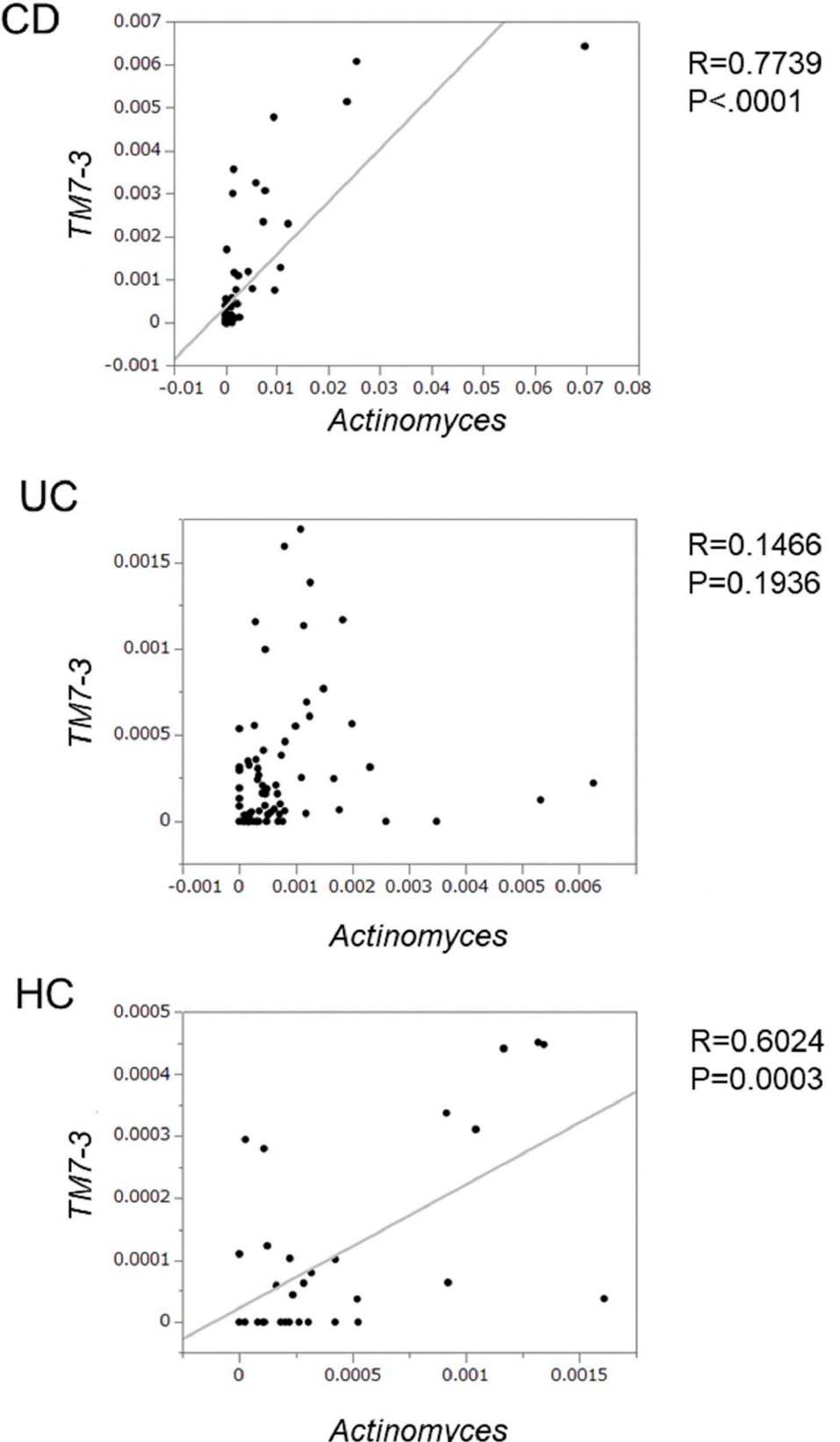

**Fig 5. Correlation of *Actinomyces* and *TM7-3* in the IBD patients and HCs.** Both the vertical and horizontal axes show the relative proportions. A p value of less than 0.05 was considered significant.

**A**

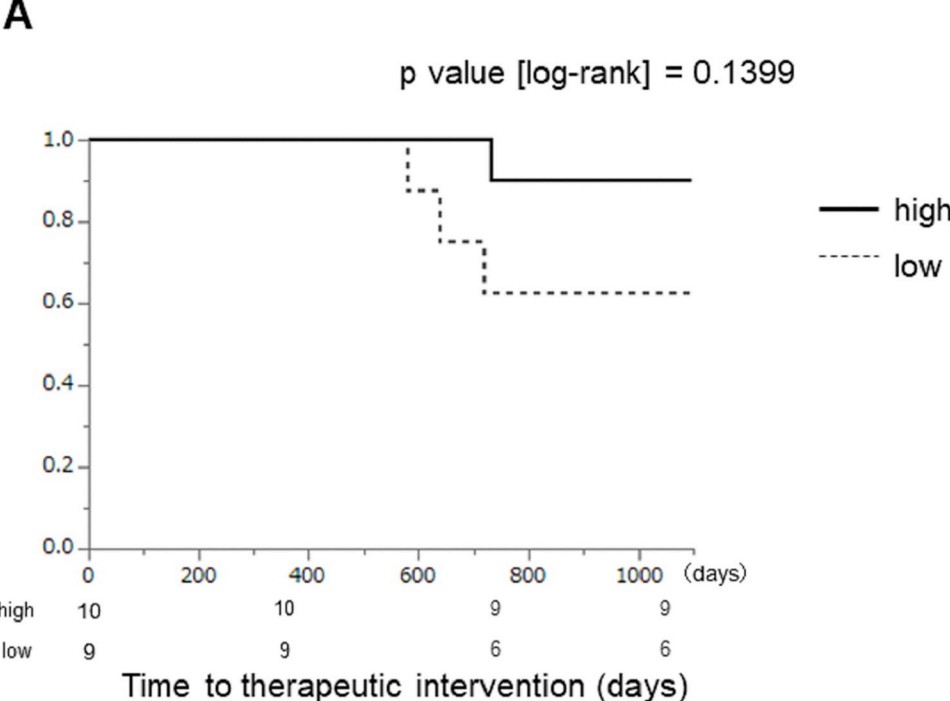

**B**

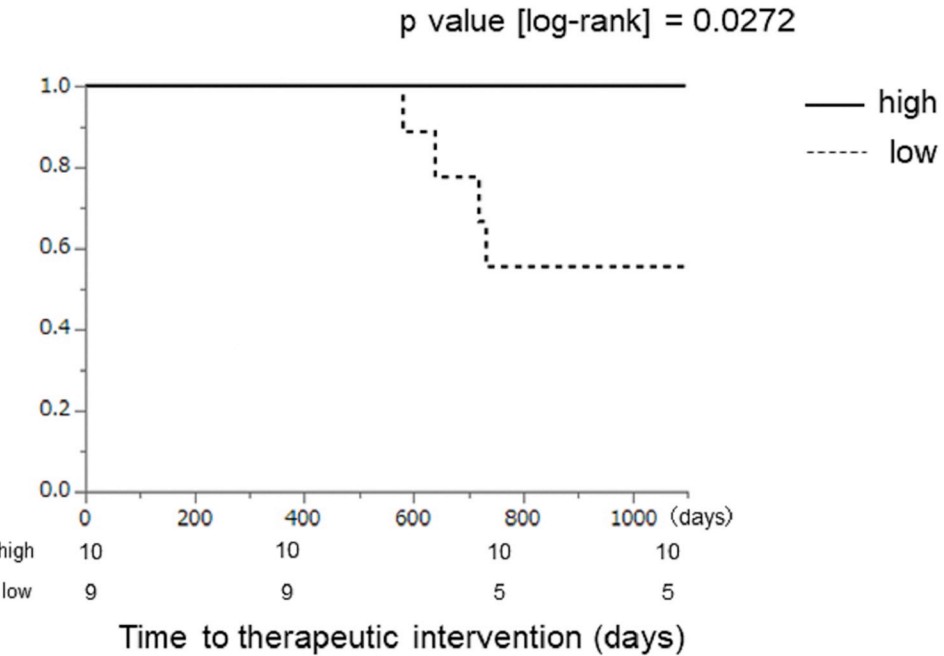

**Fig 6. Analysis of the time to therapeutic intervention for 3 years after specimen collection in CD patients.** Patients were divided into the high and the low groups by the median value of the difference of abundance in *Actinomyces* (A) or *TM7-3* (B) in the autumn compared with that in the spring. A p value of less than 0.05 was considered significant.

composition of the HCs did not fluctuate throughout the year. In this study, all participants were Japanese, and those who used antibiotics were excluded from all groups. In hunter-gatherer societies, such as the Inuit of the Canadian Arctic, no seasonal changes in the gut microbiota were observed, which was reported to be due to a westernized diet [16]. Similar to our results in healthy Japanese controls, there were no seasonal changes in the intestinal microbiota.

*Actinomyces*, a member of *Actinobacteria*, is a bacterium indigenous to the oral cavity and digestive tract and can be a causative agent of periodontitis and abdominal actinomycosis [12]. Previous reports showed that *Actinomyces* infection in IBD patients mimics fistulizing CD, tuberculosis, malignancy, or nocardiosis, and often results in the formation of a cold abscess [17,18]. There is a possibility that oral-derived bacteria can colonize the intestine and are involved in the occurrence of IBD via hydrogen sulphide production [19]. In a healthy state, oral bacteria are inhibited from entering the intestinal tract by gastric juices, bile, and the barrier function of the intestinal mucosa [20]. However, in IBD patients, the resistance of colonic fixation of foreign bacteria is reduced by oral inflammation such as periodontitis, decreased gastric mucosal barrier due to PPI use, decreased antimicrobial factors due to decreased bile acid secretion, and dysbiosis in the colon [21–23]. Common risk factors for infection by *Actinomyces* are recent dental procedures or a history of smoking and alcohol use [24]. However, it is difficult to examine whether there is a seasonal variation in the amount of smoking and alcohol use in this group because there are no data.

*TM7* is a recently described subgroup of Gram-positive uncultivable bacteria associated with oral inflammation [25]. *Saccharibacteria* was formerly known as *TM7*, which is partly composed of normal oral microbiota and is well studied due to its association with multiple mucosal diseases, such as halitosis and periodontitis [15]. There are few reports about IBD and the *TM7* phylum except for one showing that the abundance of *TM7* was increased in the gingiva of pediatric CD patients compared to HCs but decreased to the same level as that in HCs after 8 weeks of drug treatment [26]. *TM7* bacteria may play a role as promoters of inflammation in inflammatory gastrointestinal disorders with an environmental influence [27] and the increases in the abundance of *TM7* in CD patients may be associated with an inflammatory response in the intestine, but the precise mechanism has not been elucidated. Interestingly, *TM7* grows on the surface of its host bacterial species *Actinomyces odontolyticus* strain XH001 [15]. In the present study, we clearly showed that *Actinomyces* was significantly more abundant in autumn than in spring, and *TM7-3*, a member of the *TM7* phylum, was significantly more abundant in autumn than in spring and winter in the CD patients. There was a correlation between *Actinomyces* and *TM7-3* in all seasons in the CD patients but not in the UC patients. It was reported that 46% of the CD patients had dental caries compared to 25% of the HCs, possibly due to oral microbiota dysbiosis and an increased abundance of *TM7* in the mouth [28]. CD patients are more susceptible to dental caries due to nutritional deficiencies and changes in the oral microbiota, and there is less oral microbial diversity in pediatric CD patients than in HCs but not UC patients [29,30]. These findings suggest that the differences between CD and UC patients in terms of oral microbiota and small intestinal lesions may influence seasonal changes in intestinal microbiota compositions. The lack of correlation between the percentage of *Actinomyces* and *TM7-3* present in patients with UC may be due to the pathogenesis of UC itself. In addition, CD patients consume mainly carbohydrates and

limit fat contents as nutrition therapy, which can also increase dental caries and *Actinomyces* spp. in the supragingival plaque. [31] We hypothesize that the increase in *TM7* and *Actinobacteria* in the autumn may be due to a combination of factors, including changes in the diet and oral microbiota from summer to autumn. We have not been able to investigate the diet and oral circumstances in the present study; therefore, further study to investigate them in IBD patients is needed.

We initially hypothesized the increase in the abundances of *Actinomyces* and *TM7-3* in autumn in CD patients can trigger enteritis symptoms in the winter. However, interestingly, patients with increased *TM7-3* in the autumn showed better outcomes at 3 years of observation than those with no increase. Recently, Chipashvili et al. reported a protective role of *TM7* against inflammatory damage in periodontitis, suggesting that increased *TM7-3* in the gut might also inhibit the exacerbation of CD in the winter [32]. As in the reports of IBD improvement with probiotics administration, the changes in the intestinal microbiota may be involved in IBD improvement after several months [33]. There are also reports of an increase in fecal calprotectin prior to symptom exacerbations [34]. These reports support our present data that clinical symptoms possibly appear a while after disease exacerbations. And Takeuchi T, *et al.* recently revealed that acetate directs selective IgA binding to pathogenic bacteria and prevents their entry into the mucus layer of the colonic surface [35]. *Actinomyces* is an acetate-producing bacteria, and the decline of *Actinomyces* in winter may trigger an exacerbation of disease through the action of acetate, and the role of *Actinomyces* in intestinal inflammation should be further investigated. Although it is unclear why one year of observation of intestinal microbiota can assess disease course over 3 years, we speculate that seasonal changes in intestinal microbiota and some environmental factors that influence these changes may have some effect on long-term disease course. Unfortunately, we could not investigate the cause of seasonal changes in *Actinomyces* and *TM7-3*; however, there are possible explanations for the change. Changes due to the season itself (e.g. temperature and humidity) and seasonal host changes (e.g. diet and sleep) can alter the oral microbiota or directly alter the intestinal microbiota. Recently it has been reported not only do oral pathogens migrate to the intestinal tract and exacerbate periodontitis via Th17 cells, but also that Th17 cells derived from the intestinal tract migrate to the oral cavity during oral infection [36]. These seasonal changes may have influenced the fact that these two bacterial microbiotas in the stool over time are significantly higher in autumn than in spring and significantly lower in winter than in autumn. This effect may have been particularly strong in CD patients. Since only fecal samples were collected in this study, we were not able to evaluate the oral microbiota in the target population of this study, and it remains to be observed whether *TM7-3* is also increased in the oral cavity in the autumn. Further investigation is needed to elucidate mechanisms underlying the increase in *TM7-3* in the autumn only in CD and protective effect of increased *TM7-3* against CD by comparing fecal and saliva microbiota composition.

There are several limitations to this study. First, it was a single-center study and the statistical power of the study may have been affected by small sample size. Second, the patient group included outpatients with a relatively stable clinical activity index, and patients with a high activity index who required emergency surgery were excluded. Third, we have not been able to obtain information about other confounding factors that may affect the gut microbiota, such as diet, smoking, alcohol use, and recent dental treatment history. Finally, the observation period was limited to four consecutive seasons over a one-year period. Therefore, it is possible only short-term changes were captured, and it is desirable to extend the observation period in the future.

In conclusion, we explored whether *Actinomyces*, which is commonly found in the oral cavity and causes dental caries, and its symbiont, *TM7-3*, correlatively fluctuated in the faeces of

CD patients by season. The significance of seasonal changes in these bacteria in the pathogenesis of IBD needs to be further investigated.

## Supporting information

**S1 Fig. Shannon indices in the IBD patients and HCs.** The vertical axis shows the Shannon index for each season, and the horizontal axis shows the season. Bonferroni correction was performed with a paired test for correspondence. No significant difference was found.
(TIF)

**S2 Fig.** (**A**) **Comparison of relative abundances at the phylum level in the four seasons between the IBD patients and HCs.** The vertical axis shows the relative abundance, and the horizontal axis indicates each disease. The p value indicates statistical significance according to Tukey's HSD test *: $p < 0.05$, **: $p < 0.01$ (**B**) **Seasonal changes at the phylum level in the HCs.** The eight main phyla with the largest proportions were examined. Phyla are listed in alphabetical order. There was no significant seasonal variation in any of the phyla. Bonferroni correction was performed with a paired test for correspondence, and a p value less than 0.0083 was considered significant.
(TIF)

**S3 Fig. Comparison of the target bacteria in the four seasons among disease groups at the genus level.** The vertical axis shows the relative abundance, and the horizontal axis indicates each disease group. Bacteria belonging to the phyla *Actinobacteria* and *TM7* (genus level) showed seasonal variations in IBD patients. The p value indicates statistical significance according to Tukey's HSD test. * indicates $p < 0.05$.
(TIF)

**S4 Fig. Correlation with *Actinobacteria* and *TM7* in the HCs.** Both the vertical and horizontal axes show the relative abundance.
(TIF)

**S5 Fig. Correlation with *Actinomyces* and *TM7-3* in every season.** Correlations of these bacteria in the CD patients (**A**), UC patients (**B**) and HCs (**C**) are shown.
(TIF)

**S6 Fig. Correlation between the relative proportion of each bacterium and CDAI up to 1 year after the start of the study.** There was no correlation between *Actinomyces* or *TM7-3* and CDAI.
(TIF)

**S7 Fig. Seasonal changes in disease activity in CD patients.** The vertical axis shows C-reactive protein (CRP) (**A**) and CDAI (**B**), and the horizontal axis shows the season (*Sp: Spring, Su: Summer, Au: Autumn, Wi: Winter). Bonferroni correction was performed with a paired test for correspondence, and a p value of less than 0.0083 was considered significant.
(TIF)

## Acknowledgments

We wish to thank Hiroyuki Kurakami (Department of Medical Innovation, Osaka University Hospital, Suita, Osaka), a biostatistician, for his help with data processing and all participants for supporting our research. We would like to thank AJE (www.AJE.com) for English language editing.

## Author Contributions

**Conceptualization:** Mizuki Tani, Shinichiro Shinzaki, Hideki Iijima, Tetsuo Takehara.

**Data curation:** Mizuki Tani, Shinichiro Shinzaki, Daisuke Motooka, Shota Nakamura.

**Formal analysis:** Mizuki Tani, Shinichiro Shinzaki.

**Investigation:** Shinichiro Shinzaki, Akiko Asakura, Taku Tashiro, Takahiro Amano, Yuriko Otake-Kasamoto, Takeo Yoshihara, Shunsuke Yoshii, Yoshiki Tsujii, Yoshito Hayashi, Takahiro Inoue, Hideki Iijima.

**Methodology:** Shinichiro Shinzaki, Hideki Iijima.

**Supervision:** Tetsuo Takehara.

**Visualization:** Mizuki Tani, Shinichiro Shinzaki.

**Writing – original draft:** Mizuki Tani.

**Writing – review & editing:** Shinichiro Shinzaki.

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
