## [Decision Letter · Decision Letter 0]

26 Jan 2023

PONE-D-22-30672Seasonal variations in gut microbiota and disease course in patients with inflammatory bowel diseasePLOS ONE

Dear Dr. Takehara,

Thank you for submitting your manuscript to PLOS ONE. After careful consideration, we feel that it has merit but does not fully meet PLOS ONE’s publication criteria as it currently stands. Therefore, we invite you to submit a revised version of the manuscript that addresses the points raised during the review process.

We look forward to receiving your revised manuscript.

Kind regards,

Farah Al-Marzooq, MD, PhD

Academic Editor

PLOS ONE

Journal Requirements:

"The funders had no role in study design, data collection and analysis, decision to publish, or preparation of the manuscript"

4.In your Data Availability statement, you have not specified where the minimal data set underlying the results described in your manuscript can be found. PLOS defines a study's minimal data set as the underlying data used to reach the conclusions drawn in the manuscript and any additional data required to replicate the reported study findings in their entirety. All PLOS journals require that the minimal data set be made fully available. For more information about our data policy, please see http://journals.plos.org/plosone/s/data-availability.

5. Please upload a new copy of all Figures as the detail is not clear. Please follow the link for more information: https://blogs.plos.org/plos/2019/06/looking-good-tips-for-creating-your-plos-figures-graphics/" https://blogs.plos.org/plos/2019/06/looking-good-tips-for-creating-your-plos-figures-graphics/

Additional Editor Comments :

Please revise the manuscript as advised by the reviewers

Reviewers' comments:

Reviewer's Responses to Questions

**Comments to the Author**

1. Is the manuscript technically sound, and do the data support the conclusions?

Reviewer #1: Yes

Reviewer #2: Partly

2. Has the statistical analysis been performed appropriately and rigorously? 

Reviewer #1: Yes

Reviewer #2: Yes

3. Have the authors made all data underlying the findings in their manuscript fully available?

Reviewer #1: Yes

Reviewer #2: Yes

4. Is the manuscript presented in an intelligible fashion and written in standard English?

Reviewer #1: Yes

Reviewer #2: Yes

5. Review Comments to the Author

Reviewer #1: In this study, Mizuki Tani, et al. investigated the seasonality of fecal microbiota composition of inflammatory bowel disease patients and conclude that oral commensals Actinomyces and its symbiont TM7-3 fluctuated in the feces of Crohn’s disease patients by season.

Strengths of this study:

Study question is valid.

Adequate literature review was performed.

Results are clearly described.

Suggestions to consider:

- Study is limited by small sample size and including mostly patients with stable clinical activity index during the study period. Factors contributing to seasonal changes in Actinomyces and TMT-3 in CD patients is not discussed clearly.

- Did authors study other confounding factors that may affect gut microbiome like smoking, alcohol use, obesity, etc?

- Did authors notice any differences in microbiome with age and sex?

Reviewer #2: This manuscript examined the fecal microbiota of patients with inflammatory bowel disease, revealing seasonal changes in the gut microbiota. IBD is an urgent health issue, so this manuscript is timely. Overall, the manuscript is well written and the conclusions are fully supported by the findings. However, more polish should be considered. In my opinion, suitable revision should be conducted by the authors to ensure this manuscript can meet the standard quality of the PLoS One.

1. This paper mainly studies the influence of seasonal changes on intestinal flora, temperature and the diet may be the important factors, etc. Although all experimental materials were obtained in the same hospital, but they cannot represent the comparability of patients' living environment and eating habits, so it is necessary to describe the relevant background in detail.

2. The comment HV represented by the blue circle in Figure1 B should be changed to HC.

3. The format of references is not uniform, for example, in the line 556“de Vries SAG”. So all the references should be carefully verified.

4. The pictures in the full text are not clear.

5. Latin words such as Actinobacteria, etc. should be italicize.

6. The abundance of phyla Actinobacteria and TM7 in autumn was compared with spring and winter, why the summer was not mentioned?

7. Line 161-162“The median (range) ages of CD and UC 162 patients and HCs were 44 (32-55), 47 (44-57), and 33 (32-43) years, respectively.”

The age of the test group and the control group are also quite different. Do you consider the influence of age on the regulating ability of the flora, that is, the younger people have better regulating ability, so they do not get sick? Why not choose a control group of the similar age?

8. Line 284, The annotation format of references is incorrect.

9. In the full text and in the chart, p values are not written in standard at many places: line 186, p<.005 is changed to p<0.005; Change p<.0001 to p<0.0001 in line 187; Change p<.0001 to p<0.0001 in line 191; Change p<.0083 to p<0.0083 in line 220; Change p<.0083 to p<0.0083 at 228; In Fig5, change p<.0001 to p<0.0001 on the right side of figure CD; In S5A, p<.0001 is changed to p<0.0001

10. Supplementary chart missing: The chart mentioned in 250 lines of S5C Fig is missing.

6. PLOS authors have the option to publish the peer review history of their article (what does this mean?). If published, this will include your full peer review and any attached files.

Reviewer #1: No

Reviewer #2: No

---

## [Author Response · Author response to Decision Letter 0]

11 Mar 2023

Dr. Farah Al-Marzooq, MD, PhD

Academic Editor

PLOS ONE

Re: Manuscript ID: #PONE-D-22-30672

Dear Editor:

Thank you very much for your kind letter. We are very grateful for your encouraging review of our manuscript (PONE-D-22-30672), entitled “Seasonal variations in gut microbiota and disease course in patients with inflammatory bowel disease”. We have carefully considered your comments and have revised our manuscript accordingly. We have carried out additional consideration　in response to the reviewers’ suggestions. Point-by-point responses to the reviewers’ comments are provided below. All changes in the manuscript are highlighted in red. We hope that the revised manuscript is now suitable for publication in PLOS ONE.

Sincerely,

Tetsuo Takehara, M.D., Ph.D.

Osaka University Graduate School of Medicine

Department of Gastroenterology and Hepatology

takehara@gh.med.osaka-u.ac.jp

【Journal Requirements】

→Response: We have modified the manuscript and the file names to meet PLOS ONE's style requirements.

"The funders had no role in study design, data collection and analysis, decision to publish, or preparation of the manuscript"

→Response: This work was funded by the Japan Society for the Promotion of Science KAKENHI grant numbers 17H04159 and 19K08394. We have added the funding information in detail (page 22, line 445-page 23, line 446). 

→Response: We have added the approved number of the study on page 7, line106 of the revised manuscript.

4.In your Data Availability statement, you have not specified where the minimal data set underlying the results described in your manuscript can be found. PLOS defines a study's minimal data set as the underlying data used to reach the conclusions drawn in the manuscript and any additional data required to replicate the reported study findings in their entirety. All PLOS journals require that the minimal data set be made fully available. For more information about our data policy, please see http://journals.plos.org/plosone/s/data-availability.

→Response :The metagenome sequencing datasets are available in the NCBI/EMBL/DDBJ database (https://ddbj.nig.ac.jp/resource/bioproject/ PRJNA854295) The other datasets are included within the manuscript and its Supporting Information files.

5. Please upload a new copy of all Figures as the detail is not clear. Please follow the link for more information: https://blogs.plos.org/plos/2019/06/looking-good-tips-for-creating-your-plos-figures-graphics/" https://blogs.plos.org/plos/2019/06/looking-good-tips-for-creating-your-plos-figures-graphics/

→Response: We have modified and uploaded a new copy of all Figures.

→Response: We have verified that the References list is complete and accurate.

Responses to Reviewer #1

 Firstly, thank you for your review and thoughtful comments regarding our manuscript. Our responses to your comments are as follows:

- Study is limited by small sample size and including mostly patients with stable clinical activity index during the study period. Factors contributing to seasonal changes in Actinomyces and TMT-3 in CD patients is not discussed clearly.

Response: Thank you very much for your fruitful comments. We agree with the Reviewer #1’s suggestion that the small sample size may affect the results and have added this limitation on page 19, lines 381-382 of the revised manuscript. Although detailed information in this study regarding the factors contributing to seasonal changes in intestinal flora, such as diet and oral environment, we have additionally discussed the possible factors that cause seasonal changes in Actinomyces and TM7-3. We have amended the manuscript, on page 19, lines 363-367 and lines 370-374.

- Did authors study other confounding factors that may affect gut microbiome like smoking, alcohol use, obesity, etc?

Response: Thank you very much for your constructive comments. As the Reviewer #1 suggested, we examined factors that may affect the intestinal flora, such as smoking, alcohol use, and obesity. There were no significant differences in these factors between the three groups. The proportion of current smokers and drinkers was low among participants in this study, and we consider that the impact on the results is small. We have added these factors to the Table 1 (on page 10, line 168).

- Did authors notice any differences in microbiome with age and sex?

Response: We have examined the differences in intestinal flora regarding age and sex in response to the Reviewer #1’s suggestion. First, age-related difference was examined in each disease by dividing into two groups according to the median age. Actinomyces was significantly more abundant in the old group than in the young group both in CD and in HC . TM7-3 was significantly more abundant in the old group than in the young group only in CD . And seasonal variation of Actinomyces in autumn rather than spring was observed only in the old groups . 

Second, when divided into men and women, there was no difference in the proportion of abundance between men and women in any group, and there was no seasonal variation by gender. These results suggest the importance of seasonal variation of the intestinal flora especially at the old age; however, we would not like to show them in the manuscript because the sample size is too small by dividing the patients to examine the impact of gender and age on seasonal variability and we think that further research is needed. We therefore have added the description regarding the limitation in the manuscript, on page 19, lines 381-382. Thank you very much for your suggestions.

Responses to Reviewer #2

Thank you very much for your review and thoughtful comments regarding our manuscript. Our responses to your comments are as follows:

1. This paper mainly studies the influence of seasonal changes on intestinal flora, temperature and the diet may be the important factors, etc. Although all experimental materials were obtained in the same hospital, but they cannot represent the comparability of patients' living environment and eating habits, so it is necessary to describe the relevant background in detail.

Response: Thank you very much for your comments. About patients’ living environment, we have now added the information about current smoking, alcohol use, and BMI to Table 1 (on Page 10, line 168). We also agree with the Reviewer #2’s suggestion that dietary habit is one of the factors that affect intestinal flora, but unfortunately we did not obtain the data in this study, as already noted in the limitations. 

2. The comment HV represented by the blue circle in Figure1 B should be changed to HC.

Response: As suggested, the HV represented by the blue circle in Figure 1 was changed to HC.

3. The format of references is not uniform, for example, in the line 556“de Vries SAG”. So all the references should be carefully verified.

Response: As suggested, we have formatted the reference papers, from on pages 25 to 30.

4. The pictures in the full text are not clear.

Response: As mentioned, we have corrected to make it higher image.

5. Latin words such as Actinobacteria, etc. should be italicize.

Response: As suggested, the name of the intestinal microbiota has been changed to italics.

6. The abundance of phyla Actinobacteria and TM7 in autumn was compared with spring and winter, why the summer was not mentioned?

 Response: The abundance of phyla Actinobacteria and TM7 in autumn was not significantly different from summer. On the other hand, there was a significant difference between spring and autumn. We have now added the information about the abundance of phyla Actinobacteria and TM7 in summer. （page12, lines 205-206） Thank you very much for your suggestion. 

7. Line 161-162“The median (range) ages of CD and UC 162 patients and HCs were 44 (32-55), 47 (44-57), and 33 (32-43) years, respectively.”

The age of the test group and the control group are also quite different. Do you consider the influence of age on the regulating ability of the flora, that is, the younger people have better regulating ability, so they do not get sick? Why not choose a control group of the similar age? 

Response: Thank you for the important suggestion. We tried to recruit HCs to match their age with IBD patients, but since they were recruited on a volunteer basis, the age could not be completely adjusted, although the difference was not significant.

Since IBD is more common at younger ages, it is not clear whether a greater ability to regulate the intestinal flora influences the development or the regulation of the disease. We also received the age-related suggestion from the Reviewer #1, and we have examined the differences in the intestinal flora regarding age. When each group was divided into two groups by the median age, Actinomyces was significantly more abundant in the old group than in the young group both in CD and in HC . TM7-3 was significantly more abundant in the old group than in the young group only in CD . And seasonal variation of Actinomyces in autumn rather than spring was observed only in the old groups . These results suggest the importance of seasonal variation of the intestinal flora especially at the old age; but we would not like to show them in the manuscript because the sample size is too small by dividing the patients to examine the impact of age on seasonal variability and we think that further research is needed.

We have now added the information about age of HCs (page 10, lines 163-167) and the description regarding the limitation in the manuscript (page 19, lines 381-382).

8. Line 284, The annotation format of references is incorrect.

Response: As suggested, we have changed the annotation format of references.

9. In the full text and in the chart, p values are not written in standard at many places: line 186, p<.005 is changed to p<0.005; Change p<.0001 to p<0.0001 in line 187; Change p<.0001 to p<0.0001 in line 191; Change p<.0083 to p<0.0083 in line 220; Change p<.0083 to p<0.0083 at 228; In Fig5, change p<.0001 to p<0.0001 on the right side of figure CD; In S5A, p<.0001 is changed to p<0.0001

Response: As suggested, the notation of the p-value has been changed. Thank you for pointing this out.

10. Supplementary chart missing: The chart mentioned in 250 lines of S5C Fig is missing.

Response: As suggested, Supplementary Figure 5C was added. Thank you for pointing this out.

---

## [Editor Report · Decision Letter 1]

20 Mar 2023

Seasonal variations in gut microbiota and disease course in patients with inflammatory bowel disease

PONE-D-22-30672R1

Dear Dr. Takehara,

We’re pleased to inform you that your manuscript has been judged scientifically suitable for publication and will be formally accepted for publication once it meets all outstanding technical requirements.

Kind regards,

Farah Al-Marzooq, MD, PhD

Academic Editor

PLOS ONE

---

## [Editor Report · Acceptance letter]

29 Mar 2023

PONE-D-22-30672R1 

Seasonal variations in gut microbiota and disease course in patients with inflammatory bowel disease 

Dear Dr. Takehara:

I'm pleased to inform you that your manuscript has been deemed suitable for publication in PLOS ONE. Congratulations! Your manuscript is now with our production department. 

Kind regards, 

on behalf of

Dr. Farah Al-Marzooq 

Academic Editor

PLOS ONE